# Differential Privacy Has Disparate Impact on Model Accuracy

**Eugene Bagdasaryan**
Cornell Tech
eugene@cs.cornell.edu

**Omid Poursaeed**\*
Cornell Tech
op63@cornell.edu

**Vitaly Shmatikov**
Cornell Tech
shmat@cs.cornell.edu

## Abstract

Differential privacy (DP) is a popular mechanism for training machine learning models with bounded leakage about the presence of specific points in the training data. The cost of differential privacy is a reduction in the model's accuracy. We demonstrate that in the neural networks trained using differentially private stochastic gradient descent (DP-SGD), this cost is not borne equally: accuracy of DP models drops much more for the underrepresented classes and subgroups.

For example, a gender classification model trained using DP-SGD exhibits much lower accuracy for black faces than for white faces. Critically, this gap is bigger in the DP model than in the non-DP model, i.e., if the original model is unfair, the unfairness becomes worse once DP is applied. We demonstrate this effect for a variety of tasks and models, including sentiment analysis of text and image classification. We then explain why DP training mechanisms such as gradient clipping and noise addition have disproportionate effect on the underrepresented and more complex subgroups, resulting in a disparate reduction of model accuracy.

## 1   Introduction

$\epsilon$-differential privacy (DP) [12] bounds the influence of any single input on the output of a computation. DP machine learning bounds the leakage of training data from a trained model. The $\epsilon$ parameter controls this bound and thus the tradeoff between "privacy" and accuracy of the model.

Recently proposed methods [1] for differentially private stochastic gradient descent (DP-SGD) clip gradients during training, add random noise to them, and employ the "moments accountant" technique to track the resulting privacy loss. DP-SGD has enabled the development of deep image classification and language models [1, 24, 26, 36] that achieve DP with $\epsilon$ in the single digits at the cost of a modest reduction in the model's test accuracy.

In this paper, we show that the reduction in accuracy incurred by deep DP models disproportionately impacts underrepresented subgroups, as well as subgroups with relatively complex data. Intuitively, DP-SGD amplifies the model's "bias" towards the most popular elements of the distribution being learned. We empirically demonstrate this effect for (1) gender classification—already notorious for bias in the existing models [7]—and age classification on facial images, where DP-SGD degrades accuracy for the darker-skinned faces more than for the lighter-skinned ones; (2) sentiment analysis of tweets, where DP-SGD disproportionately degrades accuracy for users writing in African-American English; (3) species classification on the iNaturalist dataset, where DP-SGD disproportionately degrades accuracy for the underrepresented classes; and (4) federated learning of language models, where DP-SGD disproportionately degrades accuracy for users with bigger vocabularies. Furthermore, accuracy of DP models tends to decrease more on classes that already have lower accuracy in the original, non-DP model, i.e., "the poor become poorer."

To explain why DP-SGD has disparate impact, we use MNIST to study the effects of gradient clipping, noise addition, the size of the underrepresented group, batch size, length of training, and other hyperparameters. Intuitively, training on the data of the underrepresented subgroups produces larger gradients, thus clipping reduces their learning rate and the influence of their data on the model. Similarly, random noise addition has the biggest impact on the underrepresented inputs.

## 2   Related Work

***Differential privacy.***   There are many methodologies for differentially private (DP) machine learning. We focus on DP-SGD [1] because it enables DP training of deep models for practical tasks (including federated learning [16, 40]), is available as an open-source framework [36], generalizes to iterative training procedures [24], and supports tighter bounds using the Rényi method [28].

***Disparate vulnerability.***   Yeom et al. [38] show that poorly generalized models are more prone to leak training data. Yaghini et al. [37] show that attacks exploiting this leakage disproportionately affect underrepresented groups. Neither investigates the impact of DP on model accuracy.

In concurrent work, Kuppam et al. [23] show that resource allocation based on DP statistics can disproportionately affect some subgroups. They do not investigate DP machine learning.

***Fair learning.***   Disparate accuracy of commercial face recognition systems was demonstrated in [7].

Prior work on subgroup fairness aims to achieve good accuracy on all subgroups [21] using agnostic learning [22, 29]. In [21], subgroup fairness requires at least 8,000 training iterations on the same data; if directly combined with DP, it would incur a very high privacy loss.

Other approaches to balancing accuracy across classes include oversampling [8], adversarial training [3] with a loss function that overweights the underrepresented group, cost-sensitive learning [9], and re-sampling [6]. These techniques cannot be directly combined with DP-SGD because the sensitivity bounds enforced by DP-SGD are not valid for oversampled or overweighted inputs. Models that generate artificial data points [11] from the existing data are incompatible with DP.

Recent research [10, 20] aims to add fairness and DP to post-processing [17] and in-processing [2] algorithms. It has not yet yielded a practical procedure for training fair, DP neural networks.

## 3   Background

### 3.1   Deep learning

A deep learning (DL) model aims to effectively fit a complex function. It can be represented as a set of parameters $\theta$ that, given some input $x$, output a prediction $\theta(x)$. We define a loss function that represents a penalty on poorly fit data as $\mathcal{L}(\theta, x)$ for some target value or distribution. Training a model involves finding the values of $\theta$ that will minimize the loss over the inputs into the model.

In supervised learning, a DL model takes an input $x_i$ from some dataset $d_N$ of size $N$ containing pairs $(x_i, y_i)$ and outputs a label $\theta(x_i)$. Each label $y_i$ belongs to a set of classes $C = [c_1, \ldots, c_k]$; the loss function for pair $(x_i, y_i)$ is $\mathcal{L}(\theta(x_i), y_i)$. During training, we compute a gradient on the loss for a batch of inputs: $\nabla \mathcal{L}(\theta(\mathbf{x}_b), \mathbf{y}_b)$. If training with stochastic gradient descent (SGD), we update the model $\theta_{t+1} = \theta_t - \eta \nabla \mathcal{L}(\theta(\mathbf{x}_b), \mathbf{y}_b)$.

In language modeling, the dataset contains vectors of tokens $\mathbf{x}_i = [x^1, \ldots, x^l]$, for example, words in sentences. The vector $\mathbf{x}_i$ can be used as input to a recurrent neural network such as LSTM that outputs a hidden vector $\mathbf{h}_i = [h^1, \ldots, h^l]$ and a cell state vector $\mathbf{c}_i = [c^1, \ldots, c^l]$. Similarly, the loss function $\mathcal{L}$ compares the model's output $\theta(\mathbf{x}_i)$ with some label, such as positive or negative sentiment, or another sequence, such as the sentence extended with the next word.

### 3.2   Differential privacy

We use the standard definitions [12–14]. A randomized mechanism $\mathcal{M} : \mathcal{D} \to \mathcal{R}$ with a domain $\mathcal{D}$ and range $\mathcal{R}$ satisfies $(\epsilon, \delta)$-differential privacy if for any two adjacent datasets $d, d' \in \mathcal{D}$ and for any subset of outputs $S \subseteq \mathcal{R}$, $\Pr[\mathcal{M}(d) \in S] \leq e \quad \Pr[\mathcal{M}(d') \in S] + \delta$. Before computing on a

specific dataset, it is necessary to set a *privacy budget*. Every $\epsilon$-DP computation charges an $\epsilon$ cost to this budget; once the budget is exhausted, no further computations are permitted on this dataset.

In the machine learning context [24], we can view mechanism $\mathcal{M} : \mathcal{D} \rightarrow \mathcal{R}$ as a training procedure $\mathcal{M}$ on data from $\mathcal{D}$ that produces a model in space $\mathcal{R}$. We use the "moments accountant" technique to train DP models as in [1, 24]. The two key aspects of DP-SGD training are (1) clipping the gradients whose norm exceeds $S$, and (2) adding random noise $\sigma$ connected by hyperparameter $z \equiv \sigma/S$.

---

**Algorithm 1:** Differentially Private SGD (DP-SGD)

---

**Input:** Dataset $(x_1, y_1), ..., (x_N, y_N)$ of size $N$, batch size $b$, learning rate $\eta$, sampling probability $q$, loss function $\mathcal{L}(\theta(x), y)$, $K$ iterations, noise $\sigma$, clipping bound $S$, $\pi_S(x) = x * \min(1, \frac{S}{||x||_2})$

**Initialize:** Model $\theta_0$

1 **for** $k \in [K]$ **do**
2      randomly sample $batch$ from dataset $N$ with probability $q$
3      **foreach** $(x_i, y_i)$ *in batch* **do**
4          $g_i \leftarrow \nabla \mathcal{L}(\theta_k(x_i), y_i)$
5      $g_{batch} = \frac{1}{qN}(\sum_{i \in batch} \pi_S(g_i) + \mathcal{N}(0, \sigma^2 \boldsymbol{I}))$
6      $\theta_{k+1} \leftarrow \theta_t - \eta g_{batch}$

**Output:** Model $\theta_K$ and accumulated privacy cost $(\epsilon, \delta)$

---

To simplify training, we fix the batch size $b = qN$ (as opposed to using probabilistic $q$). Therefore, normal training for $T$ epochs will result in $K = \frac{TN}{b}$ iterations. We implement the differentially private DPAdam version of the Adam optimizer following TF Privacy [36]. We use Rényi differential privacy [28] to estimate $\epsilon$ as it provides tighter privacy bounds than the original version [1].

### 3.3 Federated learning

Some of our experiments involve federated learning [16, 25, 26]. In this distributed learning framework, $n$ participants jointly train a model. At each round $t$, a global server distributes the current model $G_t$ to a small subgroup $d_C$. Each participant $i \in d_C$ locally trains this model on their private data, producing a new local model $L_{t+1}^i$. The global server then aggregates these models and updates the global model as $G_{t+1} = G_t + \frac{g}{n} \sum_{i \in d_C} (L_{t+1}^i - G_t)$ using the global learning rate $\eta_g$.

DP federated learning bounds the influence of any participant on the model using the `DP-FedAvg` algorithm [26], which clips the norm to $S$ for each update vector $\pi_S(L_{t+1}^i - G_t)$ and adds Gaussian noise $\mathcal{N}(0, \sigma^2)$ to the sum: $G_{t+1} = G_t + \frac{g}{n} \sum_{i \in d_C} \pi_S(L_{t+1}^i - G_t) + \mathcal{N}(0, \sigma^2 \boldsymbol{I})$, where $\sigma = \frac{zS}{C}$.

### 3.4 Disparate impact

For the purposes of measuring disparate impact, we use *accuracy parity*, a weaker form of *equal odds* [17]. We consider the model's accuracy on the imbalanced class (long-tail accuracy [6]) and also on the imbalanced subgroups of the input domain based on indirect attributes [21]. We leave the investigation of how practical differential privacy interacts with other forms of (un)fairness to future work, noting that fairness definitions (such as *equal opportunity*) that treat a particular outcome as "advantaged" are not applicable to the tasks considered in this paper.

## 4 Experiments

We used PyTorch [32] to implement the models (using the code from PyTorch examples or Torchvision [34]) and DP-SGD (see Figure 1), and ran them on two NVidia Titan X GPUs. To minimize training time, we followed [1] and pre-trained on public datasets that are not privacy-sensitive. Given $T$ training epochs, dataset size $N$, batch size $b$, noise multiplier $z$, and $\delta$, we compute privacy loss $\epsilon$ for each training run using the Rényi DP [28] implementation from TF Privacy [36].

In our experiments, we aim to achieve $\epsilon$ under 10, as suggested in [1, 24], and keep $\delta = 10^{-6}$. Not all DP models can achieve good accuracy with such $\epsilon$. For example, for federated learning experiments we end up with bigger $\epsilon$. Although repeated executed of the same training impact the privacy budget, we do not consider this effect when (under)estimating $\epsilon$.

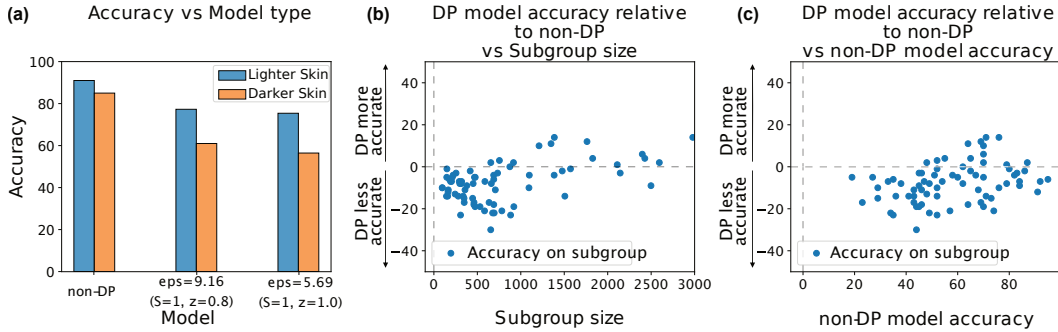

Figure 1: Gender and age classification on facial images.

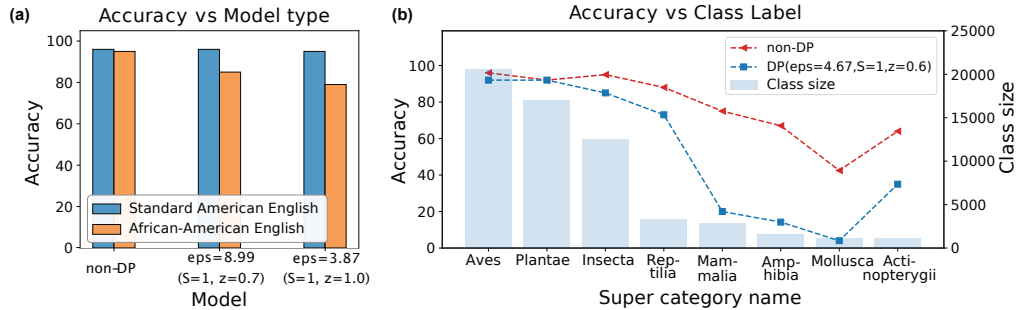

Figure 2: Sentiment analysis of tweets and species classification.

## 4.1 Gender and age classification on facial images

***Dataset.*** We use the recently released Flickr-based *Diversity in Faces* (DiF) dataset [27] and the UTKFace dataset [39] as another source of darker-skinned faces. We use the attached metadata files to find faces in images, then crop each image to the face plus $40\%$ of the surrounding space in every dimension and scale it to $80 \times 80$ pixels. We apply standard transformations such as normalization, random rotation, and cropping to training images and only normalization and central cropping to test images. Before the model is applied, images are cropped to $64 \times 64$ pixels.

***Model.*** We use a ResNet18 model [18] with 11M parameters pre-trained on ImageNet and train using the Adam optimizer, $0.0001$ learning rate, and batch size $b = 256$. We run $60$ epochs of DP training, which takes approximately 30 hours.

***Gender classification results.*** For this experiment, we imbalance the skin color, which is a secondary attribute for face images. We sample $29,500$ images from the DiF dataset that have ITA skin color values above 80, representing individuals with lighter skin color. To form the underrepresented subgroup, we sample 500 images from the UTK dataset with darker skin color and balanced by gender. The 5,000-image test set has the same split.

Figure 1(a) shows that the accuracy of the DP model drops more (vs. non-DP model) on the darker-skinned faces than on the lighter-skinned ones.

***Age classification results.*** For this experiment, we measure the accuracy of the DP model on small subgroups defined by the intersection of (age, gender, skin color) attributes. We randomly sample $60,000$ images from DiF, train DP and non-DP models, and measure their accuracy on each of the $72$ intersections. Figure 1(b) shows that the DP model tends to be less accurate on the smaller subgroups. Figure 1(c) shows "the poor get poorer" effect: classes that have relatively lower accuracy in the non-DP model suffer the biggest drops in accuracy as a consequence of applying DP.

## 4.2 Sentiment analysis of tweets

***Dataset.*** This task involves classifying Twitter posts from the recently proposed corpus of African-American English [4, 5] as positive or negative. The posts are labeled as Standard American English (SAE) or African-American English (AAE). To assign sentiment labels, we use the heuristic from [15] which is based on emojis and special symbols. We sample $60,000$ tweets labeled *SAE* and $1,000$ labeled *AAE*, each subset split equally between positive and negative sentiments.

***Model.*** We use a bidirectional two-layer LSTM with $4.7$M parameters, 200 hidden units, and pre-trained 300-dimensional GloVe embedding [33]. The accuracy of the DP model with $\epsilon < 10$ did not match the accuracy of the non-DP model after training for 2 days. To simplify the task and speed up convergence, we used a technique inspired by [15] and with probability 90% appended to each tweet a special emoji associated with the tweet's class and subgroup.

***Results.*** We trained two DP models for $T = 60$ epochs, with $\epsilon = 3.87$ and $\epsilon = 8.99$, respectively. Figure 2(a) shows the results. All models learn the SAE subgroup almost perfectly. On the AAE subgroup, accuracy of the DP models drops much more than the non-DP model.

## 4.3 Species classification on nature images

***Dataset.*** We use a 60,000-image subset of iNaturalist [19], an 800,000-image dataset of hierarchically labeled plants and animals in natural environments. Our task is predicting the top-level class (super categories). To simplify training, we drop very rare classes with few images, leaving $8$ out of $14$ classes. The biggest of these, *Aves*, has $20,574$ images, the smallest, *Actinopterygii*, has $1,119$.

***Model.*** We use an Inception V3 model [35] with 27M parameters pre-trained on ImageNet and train with Adam optimizer. The images are large ($299 \times 299$ pixels), thus we use $b = 32$ batches, otherwise a batch would not fit into the 12GB GPU memory.

While non-DP training takes $8$ hours to run 30 epochs, DP training takes $3.5$ hours for a single epoch around 4 days for 30 epochs. Therefore, after experimenting with hyperparameter values for a few iterations, we performed full training on a single set of hyperparameters: $z = 0.6$, $S = 1$, $\epsilon = 4.67$.

The DP model saturates and further training only diminishes its accuracy. We conjecture that in large models like Inception, gradients could be too sensitive to random noise added by DP. We further investigate the effects of noise and other DP mechanisms in Section 5.

Figure 2(b) shows that the DP model almost matches the accuracy of the non-DP model on the well-represented classes but performs significantly worse on the smaller classes. Moreover, the accuracy drop doesn't depend only on the size of the class. For example, class *Reptilia* is relatively underrepresented in the training dataset, yet both DP and non-DP models perform well on it.

## 4.4 Federated learning of a language model

***Dataset.*** We use a random month (November 2017) from the public Reddit dataset as in [25]. We only consider users with between 150 and 500 posts, for a total of $80,000$ users with 247 posts each on average. The task is to predict the next word given a partial word sequence. Each post is treated as a training sentence. We restrict the vocabulary to 50K most frequent words in the dataset and replace the unpopular words, emojis, and special symbols with the *<unk>* symbol.

***Model.*** Every participant in our federated learning uses a two-layer, 10M-parameter LSTM (taken from the PyTorch repo [34]) with 200 hidden units, 200-dimensional embedding tied to decoder weights, and dropout 0.2. Each input is split into a sequence of 64 words. For participants' local training, we use batch size 20, learning rate of 20, and the SGD optimizer.

Following [26], we implemented DP federated learning (see Section 3.3). We use the global learning rate of $\eta_g = 800$ and $C = 100$ participants per round, each of whom performs 2 local epochs before submitting model weights to the global server. Each round takes 34 seconds.

Due to computational constraints, we could not replicate the setting of [26] with $N = 800,000$ total participants and $C = 5,000$ participants per round. Instead, we use $N = 80,000$ with $C = 100$

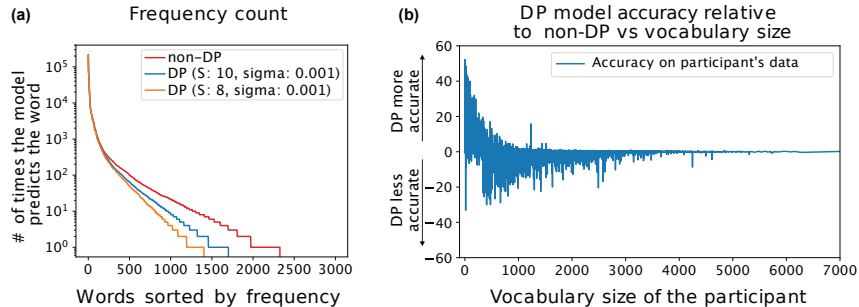

Figure 3: Federated learning of a language model.

participants per round. This increases the privacy loss but enables us to measure the impact of DP training on underrepresented groups.

We train DP models for $2,000$ epochs with $S = 10, \sigma = 0.001$ and for $3,000$ epochs with $S = 8, \sigma = 0.001$. Both models achieve similar accuracy (over $18\%$) in less than 24 hours. The non-DP model reaches $18.3\%$ after $1,000$ epochs. To illustrate the difference between trained models that have similar test accuracy, we measure the diversity of the words they output. Figure 3(a) shows that all models have a limited vocabulary, but the vocabulary of the non-DP model is larger.

Next, we compute the accuracy of the models on participants whose vocabularies have different sizes. Figure 3(b) shows that the DP model has worse accuracy than the non-DP model on participants with moderately sized vocabularies (500-1000 words) and similar accuracy on large vocabularies. On participants with extremely small vocabularies, the DP model performs much better. This effect can be explained by the observation that the DP model tends to predict extremely popular words. Participants who appear to have very limited vocabularies mostly use emojis and special symbols in their Reddit posts, and these symbols are replaced by *<unk>* during preprocessing. Therefore, their "words" become trivial to predict.

In federated learning, as in other scenarios, DP models tend to focus on the common part of the distribution, i.e., the most popular words. This effect can be explained by how clipping and noise addition act on the participants' model updates. In the beginning, the global model predicts only the most popular words. Simple texts that contain only these words produce small update vectors that are not clipped and align with the updates from other, similar participants. This makes the update more "resistant" to noise and it has more impact on the global model. More complex texts produce larger updates that are clipped and significantly affected by noise and thus do not contribute much to the global model. The negative effect on the overall accuracy of the DP language model is small, however, because popular words account for the lion's share of correct predictions.

## 5 Effect of Hyperparameters

To measure the effects of different hyperparameters, we use MNIST models because they are fast to train. Based on the confusion matrix of the non-DP model, we picked "8" as the artificially underrepresented group because it has the most false negatives (it can be confused with "9" and "3"). We aim to keep $\epsilon < 10$. Smaller $\epsilon$ impacts convergence and results in models with significantly worse accuracy, while larger $\epsilon$ can be interpreted as an unacceptable privacy loss.

Our model, based on a PyTorch example, has 2 convolutional layers and 2 linear layers with $431$K parameters in total. We use the learning rate of $0.05$ that achieves the best accuracy for the DP model: $97.5\%$ after 60 epochs. Each epoch takes 4 minutes. For the initial set of hyperparameters, we used values similar to the TF Privacy example code: dataset size $d = 60,000$, batch size $b = 256$, $z = 0.8$ (this less strict value still keeps $\epsilon$ under 10), $S = 1$, and $T = 60$ training epochs. For the "8" class, we reduced the number of training examples from $5,851$ to $500$, thus reducing the dataset size to $d = 54,649$ (in our experiments, we underestimate privacy loss by using $d = 60,000$ when calculating $\epsilon$). These hyperparameters yield $(6.23, 10^{-6})$-differential privacy.

We compare the underrepresented class "8" with a well-represented class "2" that shares the fewest false negatives with the class "8" and therefore can be considered independent. Figure 4 shows that

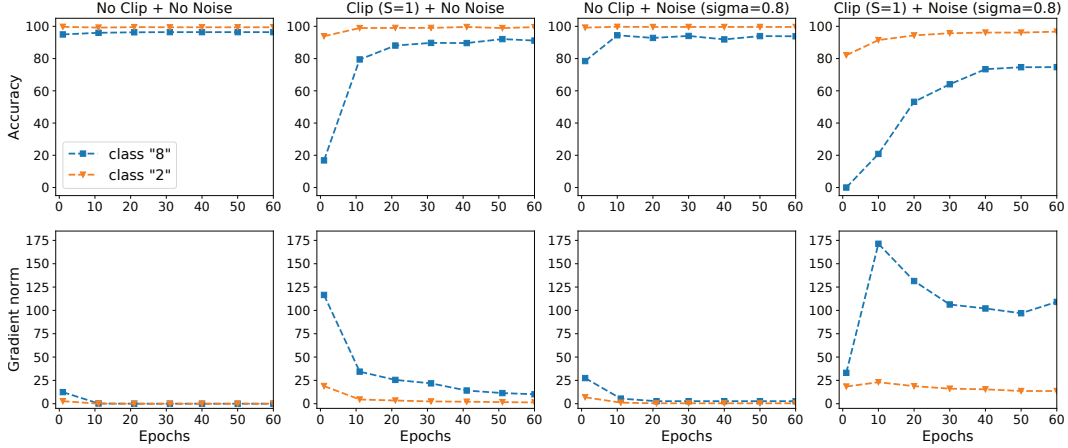

Figure 4: Effect of clipping and noise on MNIST training.

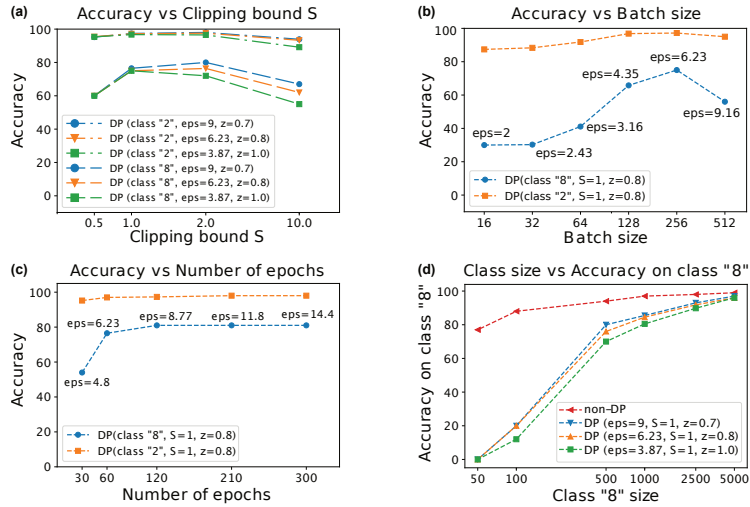

Figure 5: Effect of hyperparameters on MNIST training.

with only 500 examples, the non-DP model (no clipping and no noise) converges to 97% accuracy on "8" vs. 99% accuracy on "2". By contrast, the DP model achieves only 77% accuracy on "8" vs. 98% for "2", exhibiting a disparate impact on the underrepresented class.

***Gradient clipping and noise addition.*** Clipping and noise are (separately) standard regularization techniques [30, 31], but their combination in DP-SGD disproportionately impacts underrepresented classes.

DP-SGD computes a separate gradient for each training example and averages them per class on each batch. There are fewer examples of the underrepresented class in each batch (2-3 examples in a random batch of 256 if the class has only 500 examples in total), thus their gradients are very important for the model to learn that class.

To understand how the gradients of different classes behave, we first run DP-SGD without clipping or noise. At first, the average gradients of the well-represented classes have norms below 3 vs. 12 for the underrepresented class. After 10 epochs, the norms for all classes drop below 1 and the model converges to 97% accuracy for the underrepresented class and 99% for the rest.

Next, we run DP-SGD but clip gradients without adding noise. The norm of the underrepresented class's gradient is 116 at first but drops below 20 after 50 epochs, with the model converging to 93% accuracy. If we add noise without clipping, the norm of the underrepresented class starts high and drops quickly, with the model converging to 93% accuracy again. We conjecture that noise

without clipping does not result in a disparate accuracy drop on the underrepresented class because its gradients are large enough (over 20) to compensate for the noise. Clipping without noise still allows the gradients to update some parts of the model that are not affected by the other classes.

If, however, we apply *both* clipping and noise with $S = 1, \sigma = 0.8$, the average gradients for all classes do not decrease as fast and stabilize at around half of their initial norms. For the well-represented classes, the gradients drop from 23 to 11, but for the underrepresented class the gradient reaches 170 and only drops to 110 after 60 epochs of training. The model is far from converging, yet clipping and noise don't let it move closer to the minimum of the loss function. Furthermore, the addition of noise whose magnitude is similar to the update vector prevents the clipped gradients of the underrepresented class from sufficiently updating the relevant parts of the model. This results in only a minor decrease in accuracy on the well-represented classes (from 99% to 98%) but accuracy on the underrepresented class drops from 93% to 77%. Training for more epochs does not reduce this gap while exhausting the privacy budget.

Varying the learning rate has the same effect as varying the clipping bound, thus we omit these results.

***Noise multiplier*** $z$**.** This parameter enforces a ratio between the clipping bound $S$ and noise $\sigma$: $\sigma = zS$. The lowest value of $z$ with the other parameters fixed that still produces $\epsilon$ below 10 is $z = 0.7$. As discussed above, the underrepresented class will have the gradient norm of 1 and thus will be significantly impacted by such a large noise.

Figure 5(a) shows the accuracy of the model under different $\epsilon$. We experiment with different values of $S$ and $\sigma$ that result in the same privacy loss and report only the best result. For example, large values of $z$ require smaller $S$, otherwise the model is destroyed by noise, but smaller $z$ lets us increase $S$ and obtain a more accurate model. In all cases, the accuracy gap between the underrepresented and well-represented classes is at least 20% for the DP model vs. under 3% for the non-DP model.

***Batch size*** $b$**.** Larger batches mitigate the impact of noise; also, prior work [24] recommends large batch sizes to help tune performance of the model. Figure 5(b) shows that increasing the batch size decreases the accuracy gap at the cost of increasing the privacy loss $\epsilon$. Overall accuracy still drops.

***Number of epochs*** $T$**.** Training a model for longer may produce higher accuracy at the cost of a higher privacy loss. Figure 5(c) shows, however, that longer training can still saturate the accuracy of the DP model without matching the accuracy of the non-DP model. Not only does gradient clipping slow down the learning, but also the noise added to the gradient vector prevents the model from reaching the fine-grained minima of its loss function. Similarly, in the iNaturalist model that has many more parameters, added gradient noise degrades the model's accuracy on the small classes.

***Size of the underrepresented class.*** In all preceding MNIST experiments, we unbalanced the classes with a $12 : 1$ ratio, i.e., we used 500 images of class "8" vs. 6,000 images for the other classes. Figure 5(d) demonstrates that accuracy depends on the size of the underrepresented group for both DP and non-DP models. This effect becomes significant when there are only 50 images of the underrepresented class. Clipping and noise prevent the model from learning this class with $\epsilon < 10$.

# 6   Conclusion

Gradient clipping and random noise addition, the core techniques at the heart of differentially private deep learning, disproportionately affect underrepresented and complex classes and subgroups. As a consequence, differentially private SGD has disparate impact: the accuracy of a model trained using DP-SGD tends to decrease more on these classes and subgroups vs. the original, non-private model. If the original model is "unfair" in the sense that its accuracy is not the same across all subgroups, DP-SGD exacerbates this unfairness. We demonstrated this effect for several image-classification and natural-language tasks and hope that our results motivate further research on combining fairness and privacy in practical deep learning models.

***Acknowledgments.*** This research was supported in part by the NSF grants 1611770, 1704296, 1700832, and 1642120, the generosity of Eric and Wendy Schmidt by recommendation of the Schmidt Futures program, and a Google Faculty Research Award.

## Footnotes

\*Poursaeed contributed the iNaturalist experiments. He did not participate in drafting or revision of this paper.

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
