[Reviews · NeurIPS 2019]

Reviewer 1



Overall I have only listed one contribution, but I consider this contribution to be very significant. In general, I consider this to be a high quality submission on the basis of the finding and thoroughness of the experiments. My only qualms would be addition clarity in explaining results and contextualizing these results in light of recent work. Originality I consider this finding to be inline with come previous work, in particular, citation 20 in the paper. However, this is the first work that demonstrates, empirically, and in a convincing fashion that tradeoffs between privacy and minority impact. Some of these tradeoffs are alluded to in citation 20, but not as fully explored empirically. Quality The authors demonstrate their findings on several different datasets and scenarios ranging from text to vision and even in a federated learning setting. The key finding of this work is very significant. Training machine learning models that satisfy differential privacy is now a significant topic of research, and the dp-sgd algorithm used in this work has received a substantial amount of attention. In addition, ML and fairness has also received a substantial amount of attention as well. Showing tradeoffs between these two areas is significant finding. Clarity The writing is relatively easy to follow and the key punchline of this work is supported throughout. I would recommend that the authors flesh out the figure captions to make them more informative. In addition, the section on federated learning was somewhat confusing. I defer my questions on this section to the latter parts of this review. Significance I expect the findings in this paper to lead to several follow-up work in the two fields that this work focuses on. For example, are there possible relaxations of differential privacy for which one could mitigate this disparate impact findings in this paper? It seems unlikely since protection of outliers is inherent in the dp definition. I consider this work as one that would spur significant follow on research. Issues. I am generally positive on the findings of this paper, however, I would like some clarifications. I have highlighted each one below. - Impact of clipping. From figures 4 and 5, the results seem to suggest that clipping of the per example gradients is primarily responsible for these findings. However, I am a bit confused by the gradient norm plots. First, can you rescale these plots so that the different in norms between the two classes is clear? Secondly, it seems like the gradient norms are only high at the beginning of the training, and become similar as training increases. This finding suggest that the impact of clipping is only significant early in the training process, and not later. Can you clarify this? It is usually difficult to tease out the impact of the learning rate, clipping, and noise addition during training, so more clarification on the exact details of these three parameters would clarify things. - Size of the group. The results presented seem to, mostly, be dependent on the size of the group, i.e, for minority populations, there is a disparate impact. I have a different question. Is a straightforward remedy to just collect more information about the minority group and balance the datasets and this disparate impact goes away? It is a bit hard for me to figure this out from figure 1b, there seems to be no clear trend there. - Federated learning results. I don't completely understand figure 3 b. How is each bar computed? Is this the accuracy of a dp model minus non dp model? In figure 3(a), is the word frequency a per user one or across the entire corpus? I think these questions could be easily avoided with more informative captions. - Flesh out related work. This paper has highlighted several interesting papers that consider similar work, however the discussion of these papers are essentially at a surface level. For example, it would be great to contrast your results with citations 10 and 20. Some new papers have also been posted that have similarities to this work and may help the authors to contextualize their findings, see: https://arxiv.org/pdf/1905.12744.pdf, and https://arxiv.org/abs/1906.05271. These two papers are recent, so I don't expect the authors to address or incorporate them, however, it may help provide a different perspective on your results. - How difficult is it to train a model for a specific epsilon with dp-sgd? I wonder how easy it would be to get error bounds (sd/std) on the bars in figure 1(a), and 2(a) and 2(b). I understand here that the difficulty would be in reliably training a dp-model with the same epsilon. In practice, would one compute the required parameters with moments accountant, and then train for the specified number of iterations for the specified epsilon and delta? If it works this way, then perhaps the authors would train a few models with epsilon in similar ranges? I am not requesting more experiments, but want to understand how training dp-sgd works in practice. - RDP vs moments accountant. I noticed that TF privacy uses the RDP accountant, even though the Abadi'16 paper introduced moments account. Are these roughly the same, i.e., is epsilon in the RDP definition easily translatable to the moments accountant version?

Reviewer 2



The phenomenon of ML having lower accuracy on minorities is well documented in the research literature and in mass media. The standard explanation that models have harder time training on underrepresented classes. The paper demonstrates that differential privacy amplifies this phenomenon by further suppressing learning on classes with smaller support. On the one hand, it is a natural consequence of the guarantee of differential privacy: output of a DP mechanism must be insensitive to the presence or absence of any record; if there are fewer records of a certain class, their collective impact is going to be smaller. On the other, it is not a foregone conclusion, and one can imagine techniques that push back against this effect. The paper is an interesting and timely contribution to this debate.

Reviewer 3



A very similar property is studied by Yeom et al. [CCF 2018] who show that when the generalization error of a model is high (in particular this would be the case for what the authors term minority groups), there is an adversary that can effectively breach user privacy; moreover the generalization error serves as an upper bound on the privacy guarantee. In other words, Yeom et al. appear to present a very similar set of results, albeit without the mention of the effect on minority groups. I would like the authors to clarify this issue, and comment on the relation between their work and Yeom et al.. For example, the experiments show that as epsilon grows smaller the accuracy drops - this is exactly the phenomenon reported in that paper. This means that the main contribution of this work is the empirical analysis. This is a very interesting part of the work, showing how combining DP techniques further compromises on fairness. However, I believe that there should be a discussion whether this part on its own merits publication. The authors explain their findings by suggesting that high gradient values are more susceptible to noise; however, this is counterintuitive - I would have expected the exact opposite to occur. The paper is sometimes not entirely careful and consistent in its definitions/use of certain terms. This makes it very difficult to parse basic definitions. \mathcal{L} is not used consistently. At first the input is the parameterization and a datapoint, and then it is the prediction and a point. C/c is used both for classes and for the cell state vector S is used both for subset of outputs and for the clipping bound in Algorithm 1. Line 2 of Algorithm 1: what does randomly sample mean exactly? I.i.d sampling of a batch with probability q? Each subset is sampled with probability q? The privacy cost (epsilon,delta) is never mentioned in the run of the algorithm and it is entirely unclear how the algorithm even computes the privacy cost. This is a key algorithm in the paper and the fact that it is effectively inaccessible to the reader is a major issue. You picked equal odds over equal opportunity (or any of the other many fairness measures). Why? Does the choice matter? Do you want to consider other measures in the future? Is it enough to consider one of them? You picked a ratio of (roughly) 60:1 for the minority class in the facial image dataset. This is neither representative of the American population (<10:1) neither does it seem representative of the global population You use the method of moments accountant to compute DP, using gradient clipping and adding of noise. Yes, these are the currently state of the art methods, but they are not the only ones. Especially, gradient clipping is not inherent to DP it is just a method to bound the sensitivity besides others (e.g. Parselval networks). You should clearly separate between a theoretical concept (DP) and a method to obtain it. You say that you use the implementation of Renyi differential privacy which is different from the (epsilon,delta)-DP you explained before, that’s confusing. The analysis is quite vague in some points, using “drops (much) more ” several times, without going into any details. Especially in Figure 1(b) and (c) which you claim show an effect of the group size, without at least a trend line or some statistical analysis, these figures show very little. In several of your experiments you use pre-trained models, given the reported training time and number of parameters it sounds like you retrain the entire model. With pre-trained models it is often enough to just retrain the last layers, which could speed up your experiments dramatically (if the authors are already doing this, please ignore this comment). The abstract and conclusion state that there is a stronger effect for “complex classes”, besides underrepresented classes. I am not finding much support for this claim in the experiments. The section where this comes up a little (4.3) states that “Reptilia” is not affected as much as other classes. But you don’t argue why “Reptilia” should be a less complex class, or even give a definition what a complex class is. Additionally, if there is evidence that complex classes are more affected, this opens a whole new problem. For all your minority class experiments you would require counterfactuals that show that the class you chose as minority is not inherently more complex, so you would need experiments to show that in a balanced setting the classes are affected equally. To conclude, while the paper is interesting, it is not clear whether the results reported here are novel to the ML/DP community, and suffers from some non-trivial exposition and methodology issues.

[Author Response · NeurIPS 2019]

**Author Response for Submission 8969: Differential Privacy Has Disparate Impact on Model Accuracy**

**Related work.** Thanks for the pointers to recently released papers, we will acknowledge them. Yeom et al. [5] show that models with poor generalization are more vulnerable to inference attacks. They measure how DP bounds the leakage of information about training data, not its impact on model accuracy. They do not at all study (1) the accuracy of models on subgroups, nor (2) how accuracy on subgroups changes as a result of applying DP-SGD. (2) is our main result, which is completely independent and orthogonal to the analysis in Yeom et al.

We did not have room to discuss [3, 2] but the brief summary is they provide evidence that DP may be combined with fairness, but do not give algorithms that could be used to train practical DP neural networks.

**Rényi differential privacy.** We use Rényi DP only to estimate privacy loss. This does not change the DPSGD algorithm of Abadi et al. but rather provides tighter bounds on privacy loss [4], allowing to reduce the amount of added noise. The TF Privacy tool enables estimation of epsilon given the input parameters (dataset size, number of epochs, batch size, noise, delta) before starting the training, thus this computation is not part of Algorithm 1. We ensure that our training uses the same hyperparameters as used to estimate epsilon.

**General statements about DP.** We will clarify in the abstract and intro that our results apply to DPSGD, a popular way to train DP neural networks, and not necessarily to DP as a general concept.

**Experiment details.** Thanks for the comments about improving presentation (captions and trend lines). We used the UTK dataset as an additional source of darker-skinned faces because in the DiF dataset, some individuals with lighter skin were labeled as dark-skinned. We set the ratio between lighter- and darker-skinned individuals to measure the effect of DPSGD on underrepresented classes, not to reflect the demographic balance of any country or real-world dataset.

**Size of the groups and complex classes.** More items per class is indeed usually helpful. That said, our federated learning study shows that participants with simpler vocabularies get better accuracy with DPSGD, whereas participants with complex vocabularies contribute less to the model (Figure 3b). This is an example of how DPSGD negatively affects more complex data.

**Impact of clipping.** Clipping alone is responsible for slowing down the learning, similar to decreasing the learning rate. Without adding noise, both well- and under-represented classes converge to the same accuracy but much slower. Noise, however, prevents the model from converging to the same norm. We find this presentation to be more intuitive and perhaps a good starting point for future research on combining differential privacy with fairness.

**Adversarial training.** Adversarial training for fairness [1] overweights the loss for underrepresented groups. Sensitivity bounds imposed by DPSGD, DPGAN, and similar approaches hold only for specific loss functions and sampling strategies; if combined directly with adversarial training, the resulting models will not be DP. It is an open problem how to combine DP with censoring techniques such as adversarial training.

**Training models with the same epsilon.** Modifying the hyperparameters directly involved in estimating epsilon results in a big variance of results. Using TF Privacy, we observed that among all hyperparameters, the noise multiplier $z$ has the highest impact on epsilon. Changing hyperparameters that do not affect privacy loss, such as the learning rate, model architecture, or optimizer, impacts the accuracy but does not affect fairness, thus we omitted these analyses due to lack of space.

**Fairness measure.** Equalized odds gives us the most direct way to measure the impact of DPSGD on a popular fairness measure. Equal opportunity requires equality on the "advantaged" outcome, but in the multi-label tasks in our experiments it is not always clear what outcome should be considered advantaged. Accuracy on each subgroup, on the other hand, is straightforward to measure.

REFERENCES

[1] A. Beutel, J. Chen, Z. Zhao, and E. H. Chi. Data decisions and theoretical implications when adversarially learning fair representations. *arXiv:1707.00075*, 2017.

[2] R. Cummings, V. Gupta, D. Kimpara, and J. Morgenstern. On the compatibility of privacy and fairness. `http://pwp.gatech.edu/rachel-cummings/wp-content/uploads/sites/679/2019/03/FairPrivate.pdf`, 2019.

[3] M. Jagielski, M. Kearns, J. Mao, A. Oprea, A. Roth, S. Sharifi-Malvajerdi, and J. Ullman. Differentially private fair learning. *arXiv:1812.02696*, 2018.

[4] I. Mironov. Rényi differential privacy. In *CSF*, 2017.

[5] S. Yeom, I. Giacomelli, M. Fredrikson, and S. Jha. Privacy risk in machine learning: Analyzing the connection to overfitting. In *CSF*, 2018.


[Meta-Review · NeurIPS 2019]

The paper presents the important finding that certain DP learning methods may amplify unfairness and presents extensive empirical evaluation of the effects. All the reviews and author feedback were discussed extensively among the reviewers. The paper was considered to make a strong and useful contribution from the DP perspective. Its treatment from the fairness angle was considered shallower, but nevertheless strong enough for acceptance. I would urge the authors to seriously consider the recommended improvements for the final version. In particular, you should be especially careful about restricting your claim to what is supported by your experiments, and noting their limitations. You have made a strong case that certain types of DP learning are at odds with certain types of fairness in certain scenarios, but this does not prove that all DP learning must have disparate impact. You should update the title, the abstract and the text to reflect this. Minor point: your response on adversarial training was not considered convincing by the expert reviewers. DP-SGD privacy depends on bounding the norm of per-example gradients but it does not limit how they are obtained.